# Improvement in Fine Manual Dexterity in Children with Spinal Muscular Atrophy Type 2 after Nusinersen Injection: A Case Series

**DOI:** 10.3390/children8111039

**Published:** 2021-11-11

**Authors:** Minsu Gu, Hyun-Ho Kong

**Affiliations:** Department of Rehabilitation Medicine, Chungbuk National University Hospital, Cheongju 28644, Korea; msgrehab@gmail.com

**Keywords:** spinal muscular atrophy (SMA), nusinersen, fine manual dexterity

## Abstract

Although nusinersen has been demonstrated to improve motor function in patients with spinal muscular atrophy (SMA), no studies have investigated its effect on fine manual dexterity. The present study aimed to investigate the ability of nusinersen to improve fine manual dexterity in patients with SMA type 2. A total of five patients with SMA type 2 were included. The Hammersmith Functional Motor Scale (expanded version) (HFMSE) and Purdue Pegboard (PP) tests were used to evaluate gross motor function and fine manual dexterity, respectively, until 18 months after nusinersen administration. HFMSE scores improved by 3–10 points (+13–53%) in all patients following nusinersen administration. PP scores also improved in all patients, from 4 to 9 points (+80–225%) in the preferred hand and from 3 to 7 points (+60–500%) in the non-preferred hand. These results suggest that nusinersen treatment improved both gross motor function and fine manual dexterity in children with SMA type 2. Addition of the PP test may aid in evaluating the fine manual dexterity essential for activities of daily living in these patients.

## 1. Introduction

Spinal muscular atrophy (SMA) is an autosomal recessive disorder that affects the motor neurons in the anterior horn of the spinal cord, resulting in muscle atrophy and loss of muscle strength [1]. SMA is caused by insufficient production of SMN protein due to deletion or mutation of the survival motor neuron 1 (SMN 1) gene [2,3].

Recently, nusinersen targeting the SMN gene has been used as a treatment for SMA. Previous research has consistently demonstrated that nusinersen treatment improves both gross motor function as measured using the Hammersmith Functional Motor Scale (expanded version) (HFMSE) and upper extremity motor function as measured using the Revised Upper Limb Module (RULM) in patients with later-onset SMA [4].

Although the RULM also includes some items related to hand dexterity, such as picking up coins and tearing a piece of paper, it is difficult to assess quantitative changes in dexterity because the tool utilizes a three-point scale (0, 1, 2). Previous studies only addressed changes in the total RULM score following administration of nusinersen, without performing subgroup analysis for specific items [4,5]. Therefore, while such studies were able to confirm improvements in general upper limb motor function following nusinersen administration, they were unable to confirm whether patients exhibited improvements in fine manual dexterity. The present study is the first to investigate and demonstrate the effect of nusinersen on fine manual dexterity in patients with SMA type 2.

## 2. Materials and Methods

### 2.1. Patients

A total of five patients with 5q SMA, confirmed based on SMN1 genetic documentation, were included in this study. All patients had a clinical classification of SMA type 2 and received neither permanent ventilator support nor enteral feeding. Between May 2019 and December 2019, the patients were referred to the Department of Rehabilitation Medicine to evaluate functional changes before and after nusinersen administration.

Nusinersen was administered intrathecally at a dose of 12 mg on days 0 (1st), 14 (2nd), 28 (3rd), and 63 (4th) according to the protocol, following which, it was administered once every 4 months for maintenance. Functional evaluations were performed before starting nusinersen treatment, between the 3rd and 4th doses, and before administration during the maintenance period. Patients underwent follow-up for a total of 18 months.

This study was approved by the Institutional Review Board of Chungbuk National University (CBNUH 2021-08-011), who waived the requirement for informed consent due to the retrospective nature of the study.

### 2.2. Functional Assessments

The 33-item HFMSE was developed to evaluate gross motor function related to daily living in patients with SMA type 2 or 3. Each item is scored from 0 (no response) to 2 (full response), with total scores ranging from 0 to 66 [6,7].

The Purdue Pegboard (PP) test is a standardized assessment of fine manual dexterity that is mainly used to evaluate functional abnormalities in patients with neurological impairment or developmental delay. There are normative data for most age groups, as well as reference data for preschool children over the age of 2 years, 6 months [8].

The PP test was used to evaluate fine manual dexterity in each hand. The PP test assesses the patient’s ability to pick up pegs one at a time from a cup on top of the pegboard and insert them into the holes as quickly as possible. The test was first performed using the preferred hand followed by the non-preferred hand, and the number of pegs inserted within 30 s was measured [8].

The assessments were performed by one trained clinical evaluator, and training was conducted to establish reliability before data collection began.

## 3. Results

### 3.1. Baseline Characteristics

Baseline characteristics for the five patients included in the study are summarized in Table 1. All five patients were female and had SMA type 2. Genetic sequencing analysis indicated that the SMN2 gene copy number was 3 in all 5 patients. The age at the onset of SMA symptoms ranged from 12 to 14 months, while the age at SMA diagnosis ranged from 2 to 24 months. The age at initiation of nusinersen treatment ranged from 12 to 14 months (Table 1).

### 3.2. Efficacy Results

#### 3.2.1. Hammersmith Functional Motor Scale (Expanded Version)

Baseline HFMSE scores before nusinersen administration ranged from 10 to 40 points. At 18 months after nusinersen administration, HFMSE scores had improved in all patients, ranging from a minimum of +3 points to a maximum of +10 points (+13–53%). Although there were variations among patients, gross motor functions related to trunk control such as lying, rolling, sitting, crawling, and kneeling [9] tended to improve (Table 2 and Figure 1).

#### 3.2.2. Purdue Pegboard Test

In the preferred hand, PP scores before and 18 months after the initiation of nusinersen administration ranged from 2 to 7 and from 6 to 14, respectively. The PP score of the preferred hand improved in all patients, ranging from +4 to +9 (+80–225%) when compared with the baseline score (Table 3 and Figure 2a).

In the non-preferred hand, PP scores before and 18 months after the initiation of nusinersen administration ranged from 1 to 6 and from 6 to 13, respectively. The PP score of the non-preferred hand also improved in all patients, ranging from +3 to +7 (+60–500%) when compared with the baseline score (Table 3 and Figure 2b).

Shown in Figure 3, most subjects—except for patient 3—had lower PP scores than normative data of the same age and sex before nusinersen administration; however, the PP scores in both hands of patients 2, 3, and 5 after nusinersen administration (at 18 months) improved to the normal range.

## 4. Discussion

In this study, we investigated the effects of nusinersen in five patients with SMA type 2. Gross motor function as measured using the HFMSE improved in all patients after 18 months of treatment, and the functional improvements were mainly related to trunk control. Moreover, fine manual dexterity, as evaluated using the PP test, significantly improved in all patients following nusinersen treatment, and there was no difference between the preferred and non-preferred hands.

In the present study, we observed little or no improvements in standing, walking, running, or jumping ability as measured using the HFMSE; however, scores for items related to trunk control, such as lying, rolling, sitting, crawling, and kneeling improved following treatment [9]. In a previous study by Rosenblum et al., development of trunk control was identified as a prerequisite for upper extremity function and manual dexterity in healthy children [10]. Wang et al. also noted that development of trunk control in preterm infants improved fine motor skills [11]. This is because trunk stability plays an important role in upper limb motor function [12]. In accordance with previous findings, the improvements in PP score observed in this study are presumed to include the effect of improved trunk control, as measured using the HFMSE.

In addition, a recent study by Bram et al. showed significant improvements in hand grip strength and hand motor function in adult patients with SMA type 3 and 4 treated with nusinersen [5]. Similarly, administration of nusinersen may have improved fine motor function of the hand itself in our patients with SMA type 2.

In comparing the fine manual dexterity of SMA type 2 patients treated with nusinersen versus normal children, most SMA patients before receiving nusinersen had lower PP test scores than normal children of the same age and sex. However, the scores improved to the normal range observed from healthy children in most patients treated for 18 months (Figure 3) [8,13]. In a previous study, it was reported that when nusinersen was administered to infants during the pre-symptomatic stage, most patients achieved the motor milestone within the window for healthy children [14]. Similar to the results of the previous study, it is postulated that nusinersen administration rapidly improved fine manual dexterity in SMA patients; thus, they were able to reduce the gap with the normal fine motor milestone.

Previous studies have demonstrated that patients with SMA treated with nusinersen exhibit improvements not only in HFMSE scores but also in upper arm skill, as assessed using the RULM [4]. The RULM is designed to evaluate upper limb functions closely related to activities of daily living [15]. Although the RULM has the advantage of evaluating the overall function of the upper extremities, it has limitations in quantitatively measuring fine manual dexterity. Considering that most patients with SMA type 2 cannot stand alone or walk with assistance, even if they receive nusinersen treatment, fine manual dexterity is important for activities of daily living in these patients. Simply adding the PP test to the battery of existing evaluation tools may aid in providing a more detailed assessment of fine manual dexterity in patients with SMA.

Since this study was conducted only on patients with SMA type 2, the effect of nusinersen on fine manual dexterity in patients with other subtypes of SMA could not be identified. In a previous study, comparing the effects of nusinersen on SMA types 2 and 3, the change in the HFMSE score was greater in SMA type 2 than in SMA type 3 (+10.8 points versus +1.8 points) while the change in the upper limb module (ULM) score was also greater in SMA type 2 [16]. Considering the results of this previous study, it is presumed that the benefits of nusinersen on fine manual dexterity will also be greater in type 2 SMA than in other later-onset SMA types; however, this remains to be confirmed through further studies.

The present study had some limitations. Although nusinersen improved fine manual dexterity in a small number of subjects with SMA, this effect may not be statistically significant in large-scale studies. The statistical significance of the results of this study will be confirmed through a large multicenter study. Additionally, only the PP test was used to evaluate fine manual dexterity. Use of other tools to evaluate fine manual dexterity may have yielded more robust findings. Despite these limitations, this study is meaningful in that it is the first to report improvements in fine manual dexterity after nusinersen administration in patients with SMA type 2. Nonetheless, further studies including larger numbers of patients are required to verify our findings.

## 5. Conclusions

Changes in PP scores from baseline to 18 months confirmed that nusinersen treatment improved fine manual dexterity in patients with SMA type 2. Simply adding the PP test to the existing battery of evaluation tools may help to provide a more thorough assessment of the fine manual dexterity essential for daily living activities in patients with SMA type 2.

## Figures and Tables

**Figure 1 children-08-01039-f001:**
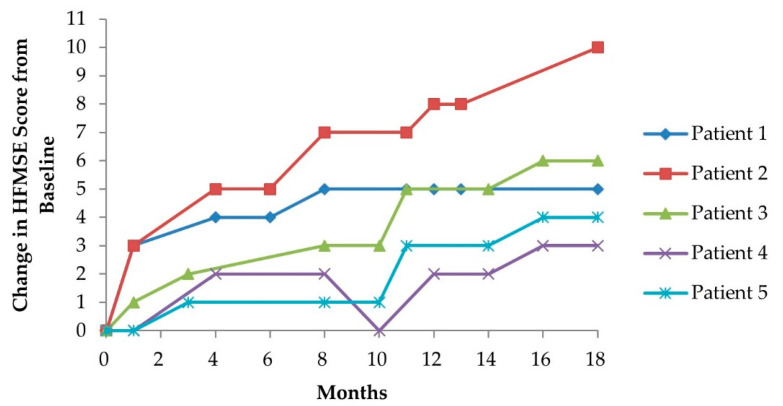
Change in HFMSE score from baseline to 18 months after nusinersen injection. HFMSE—Hammersmith Functional Motor Scale (expanded version).

**Figure 2 children-08-01039-f002:**
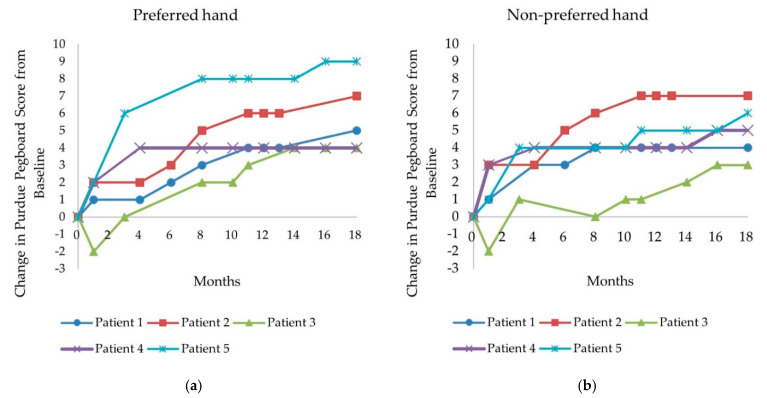
Change in Purdue Pegboard score from baseline to 18 months. (**a**) Preferred hand; (**b**) non-preferred hand.

**Figure 3 children-08-01039-f003:**
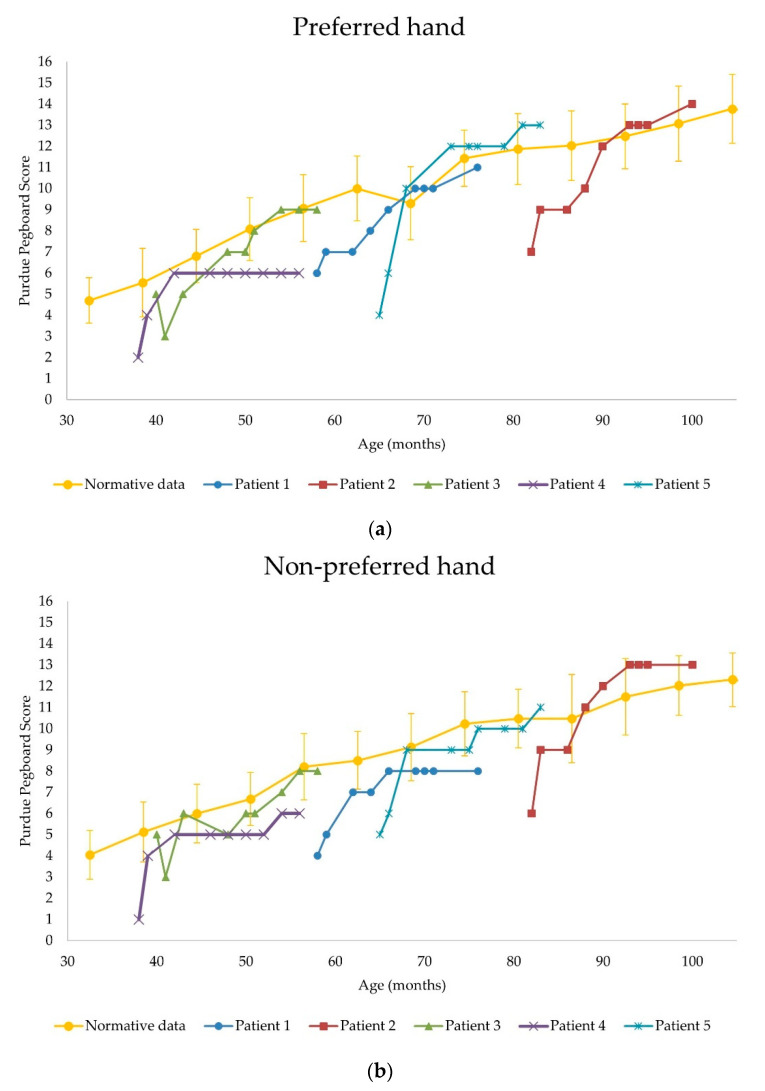
Comparison of Purdue Pegboard score changes in patients versus normative data. (**a**) Preferred hand; (**b**) non-preferred hand.

**Table 1 children-08-01039-t001:** Baseline characteristics of the study patients.

Patient Number	Sex	SMA Type	SMN2 Copy Number	Age at Symptom Onset (Month)	Age at Diagnosis (Month)	Age at First Dose (Month)
1	Female	2	3	12	2	58
2	Female	2	3	13	24	82
3	Female	2	3	14	23	40
4	Female	2	3	12	21	38
5	Female	2	3	13	21	65

SMA—spinal muscular atrophy; SMN—survival motor neuron.

**Table 2 children-08-01039-t002:** Change in HFMSE scores from baseline after nusinersen injection.

Patient Number	Baseline HFMSE Score	∆ Lying and Rolling	∆ Sitting	∆ Crawling and Kneeling	∆ Standing	∆ Walking, Running, and Jumping	HFMSE Score after 7th Dose	∆ HFMSE Score (Δ%)
1	10	+2	+3	0	0	0	15	+5 (+50%)
2	19	+5	+1	+4	0	0	29	+10 (+53%)
3	40	+3	0	+1	+1	+1	46	+6 (+15%)
4	13	0	+3	0	0	0	16	+3 (+23%)
5	30	+3	0	+1	0	0	34	+4 (+13%)

HFMSE—Hammersmith Functional Motor Scale (expanded version); ∆—amount of change.

**Table 3 children-08-01039-t003:** Change in Purdue Pegboard scores from baseline.

Patient Number	Δ HFMSE Score (Δ%)	Preferred Hand PP Score		Non-Preferred Hand PP Score	
		Baseline	After 7th Dose	Baseline	After 7th Dose
1	+5 (+50%)	6	11	4	8
Δ PP score (Δ%)		+5 (+83%)		+4 (+100%)	
2	+10 (+53%)	7	14	6	13
Δ PP score (Δ%)		+7 (+100%)		+7 (+117%)	
3	+6 (+15%)	5	9	5	8
Δ PP score (Δ%)		+4 (+80%)		+3 (+60%)	
4	+3 (+23%)	2	6	1	6
Δ PP score (Δ%)		+4 (+200%)		+5 (+500%)	
5	+4 (+13%)	4	13	5	11
Δ PP score (Δ%)		+9 (+225%)		+6 (+120%)	

HFMSE—Hammersmith Functional Motor Scale (expanded version); PP—Purdue Pegboard; ∆—amount of change.

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
