# Peer review of "Improvement in Fine Manual Dexterity in Children with Spinal Muscular Atrophy Type 2 after Nusinersen Injection: A Case Series"

_children, 2021, doi:10.3390/children8111039_

Round 1
Reviewer 1 Report
The article by Kong et al “ Improvement in ….: A case series” is well written and findings presented in this article has some meaningful facts related to the patients associated with SMA disease and who is getting Nusinersen treatment. The result suggested that newly developed drug is effective for the improvement of fine manual dexterity which is an essential feature related to activities of daily living for those affected children. Data presented in this article suggested that overall improvement after 18 months of treatment in the gross motor function; specially in l trunk control, while fine manual dexterity also significantly improved on Nusinersen treatment.
- Major limitations of this study is the sample size which is very low, and even if the data shows promising improvements in motor skills, whether these data holds these observations in large-scale studies with statistical significance is unknown. Authors should comment on this point and included in the discussion.
- Also authors only observed these result only in one SMA type, it will be interesting to see the comparable observations from other SMA types so that spectrum of Nusinersen treatment can be determined with SMA types. Authors should bring these point into the discussion with references if available. If not, they should include their insight based on their experiences. This will help many families suffering from this disease.
- All the motor skill observations was done by only one trained clinical evaluator. These clinical evaluations are important aspects of this whole study. In my opinion, these scoring based system should have been evaluated by at least two evaluators. Authors should comment why they choose only one evaluator for this important study.
- Table2- The pharase “HFMSE score after 18 months after first injection” is very confusing. It should be written clearly after which dose(e.g, after between 3 and 4 dose) scoring was performed.
- Clarification from the authors. In fig.1 for Patient 4 one data point move from score 2 to 0 between 8-10 month before coming back again to 2 between 10-12 month. This means whatever the score gained by this patient up to 8 months suddenly lost and again regained in next two month. This means the effect of Nusinersen could be reversible? Authors should explain, and if this brings important points that need to discussed, then should include in their discussion. Same observation in Fig.2 for patient 3, but in initial month. If scoring system is based on arbitrary number from 0-…., then why in figure 2, it moves to -2 scale.
Despite all limitations, I believe this article has merit which can benefit the family suffering from SMA, and bring closure to the people who have been investigating the effect of Nusinersen in SMA patient.
Author Response
Dear Reviewer,
We really appreciate the reviewer’s detailed comments on our manuscript. The reviews have raised some excellent questions, and we sought to address each issue in detail in this letter and in the attached revised manuscript. You will find the reviewer’s comments in italic, followed by our detailed response. The original text is shown in blue, and the revised or added text is shown in red.
We hope that you will find our responses to be adequate. We thank you again for your time and effort spent reading and commenting on our manuscript.
1) Major limitations of this study is the sample size which is very low, and even if the data shows promising improvements in motor skills, whether these data holds these observations in large-scale studies with statistical significance is unknown. Authors should comment on this point and included in the discussion.
: We appreciate the reviewer's comment about the most important limitation of this study. As the reviewer pointed out, although nusinersen had an effect on the improvement of fine manual dexterity in a small number of subjects with SMA, this effect may not be statistically significant in large-scale studies. However, it should be considered that SMA is a rare disease with a prevalence of around 1-2 per 100,000 persons [1] and the number of subjects treated with nusinersen is less than 30 to 80 even in the multi-country and multi-center study [2-6]. As the reviewer already knows well, a large-scale study of nusinersen is difficult to conduct due to the low prevalence of SMA and the high cost of the drug (79,000 USD per vial in Korea). Although this study is a case series with a small number of subjects, this study is meaningful as the first report on the effect of nusinersen on fine manual dexterity, which was not covered in previous studies. In addition, since most of the previous studies analyzed data together by classifying SMA types 2 and 3 as later-onset SMA [2,5,6], it was difficult to analyze the effect of nusinersen in each subtype with different levels of physical functions. However, this study could ensure more homogeneity of the study population by including only SMA type 2 who the function of the upper extremities on the sitting position is very important for the activity of daily living than other SMA subtypes because independent ambulation is impossible. The authors expect that clinicians who treat SMA patients with nusinersen will be more interested in fine manual dexterity through this study, and we are planning a large multi-center study to verify the result of this study. The part pointed out by the reviewer has been added to the discussion section.
[4. Discussion 6th paragraph]
The present study had some limitations, including its small sample size and lack of a control group. Additionally, only the PP test was used to evaluate fine manual dexterity.
à
The present study had some limitations. Although nusinersen improved fine manual dexterity in a small number of subjects with SMA, this effect may not be statistically significant in large-scale studies. The statistical significance of the results of this study will be confirmed through a large multicenter study. Additionally, only the PP test was used to evaluate fine manual dexterity.
2) Also authors only observed these result only in one SMA type, it will be interesting to see the comparable observations from other SMA types so that spectrum of Nusinersen treatment can be determined with SMA types. Authors should bring these point into the discussion with references if available. If not, they should include their insight based on their experiences. This will help many families suffering from this disease.
: We appreciate the reviewer's detailed comment. As the reviewer pointed out, the value of this study would have been higher if the effects of nusinersen on several SMA subtypes were compared. Although a thorough literature review was conducted, there was no study that analyzed the effect of nusinersen on fine manual dexterity by subtype of SMA. However, in a previous study comparing the effects of nusinersen on SMA types 2 and 3 (over 3 years), the change in Hammersmith Functional Motor Scale-Expanded version (HFMSE) was greater in SMA type 2 than SMA type 3 (SMA type 2 vs. 3, +10.8 points vs. +1.8 pints), and the change in Upper Limb Module (ULM) was also greater in SMA type 2 [2]. Considering the results of the previous study, it is presumed that improvement in fine manual dexterity through nusinersen administration will also be greater in SMA type 2 compared to other later-onset SMA types. Related text has been added to the discussion section.
[4. Discussion (added after 5th paragraph)]
Since this study was conducted only on patients with SMA type 2, the effect of nusinersen on fine manual dexterity in patients with other subtypes of SMA could not be identified. In a previous study comparing the effects of nusinersen on SMA types 2 and 3, the change in the HFMSE score was greater in SMA type 2 than in SMA type 3 (+10.8 points vs. +1.8 points) while the change in the Upper Limb Module (ULM) score was also greater in SMA type 2. Considering the results of this previous study, it is presumed that the benefits of nusinersen on fine manual dexterity will also be greater in type 2 SMA than in other later-onset SMA types; however, this remains to be confirmed through further studies.
3) All the motor skill observations was done by only one trained clinical evaluator. These clinical evaluations are important aspects of this whole study. In my opinion, these scoring based system should have been evaluated by at least two evaluators. Authors should comment why they choose only one evaluator for this important study.
: We really appreciate your great opinion. Following the reviewer's opinion, if at least two evaluators conducted the test on the same patient and checked inter-rater reliability among examiners, it would have been possible to present a more definite result. However, the Purdue Pegboard (PP) test, developed in 1948 to measure the fine motor function of the upper extremities, is a standardized measurement tool which its validity and reliability (test-retest reliability, inter-rater reliability, etc.) have been verified through a large number of previous studies [7-9]. In addition, the PP test was performed according to the manual of the test, so even if one trained clinical evaluator performed the test, we consider the data from this study are reliable.
4) Table2- The phrase “HFMSE score after 18 months after first injection” is very confusing. It should be written clearly after which dose (e.g, after between 3 and 4 dose) scoring was performed.
: We appreciate your detailed comment. When nusinersen is administered according to the protocol, 18 months after the first administration is after the 7th dose and before the 8th dose. All patients in this study were administered according to the protocol, so the phrases “HFMSE score at 18 months after first injection” in Table 2 and “18 months after nusinersen injection” in Table 3 were modified.
[Table 2]
HFMSE score at 18 months after first injection
à HFMSE score after 7th dose
Table 2. Change in HFMSE scores from baseline after nusinersen injection
|
Patient Number |
Baseline HFMSE score |
∆ Lying & Rolling |
∆ Sitting |
∆ Crawling & Kneeling |
∆ Standing |
∆ Walking, Running, & Jumping |
HFMSE score after 7th dose |
∆ HFMSE Score (Δ%) |
|
1 |
10 |
+ 2 |
+ 3 |
0 |
0 |
0 |
15 |
+ 5 (+ 50%) |
|
2 |
19 |
+ 5 |
+ 1 |
+ 4 |
0 |
0 |
29 |
+ 10 (+ 53%) |
|
3 |
40 |
+ 3 |
0 |
+ 1 |
+ 1 |
+ 1 |
46 |
+ 6 (+ 15%) |
|
4 |
13 |
0 |
+ 3 |
0 |
0 |
0 |
16 |
+ 3 (+ 23%) |
|
5 |
30 |
+ 3 |
0 |
+ 1 |
0 |
0 |
34 |
+ 4 (+ 13%) |
[Table 3]
18 months after nusinersen injection
à After 7th dose
Table 3. Change in Purdue Pegboard scores from baseline.
|
Patient number |
Δ HFMSE score (Δ%) |
Preferred hand PP score |
Non-Preferred hand PP score |
||
|
Baseline |
After 7th dose |
Baseline |
After 7th dose |
||
|
1 |
+ 5 (+ 50%) |
6 |
11 |
4 |
8 |
|
Δ PP score (Δ%) |
|
+ 5 (+ 83%) |
+ 4 (+ 100%) |
||
|
2 |
+ 10 (+ 53%) |
7 |
14 |
6 |
13 |
|
Δ PP score (Δ%) |
|
+ 7 (+ 100%) |
+ 7 (+ 117%) |
||
|
3 |
+ 6 (+ 15%) |
5 |
9 |
5 |
8 |
|
Δ PP score (Δ%) |
|
+ 4 (+ 80%) |
+ 3 (+ 60%) |
||
|
4 |
+ 3 (+ 23%) |
2 |
6 |
1 |
6 |
|
Δ PP score (Δ%) |
|
+ 4 (+ 200%) |
+ 5 (+ 500%) |
||
|
5 |
+ 4 (+ 13%) |
4 |
13 |
5 |
11 |
|
Δ PP score (Δ%) |
|
+ 9 (+ 225%) |
+ 6 (+ 120%) |
||
5) Clarification from the authors. In fig.1 for Patient 4 one data point move from score 2 to 0 between 8-10 month before coming back again to 2 between 10-12 month. This means whatever the score gained by this patient up to 8 months suddenly lost and again regained in next two month. This means the effect of Nusinersen could be reversible? Authors should explain, and if this brings important points that need to discussed, then should include in their discussion. Same observation in Fig.2 for patient 3, but in initial month. If scoring system is based on arbitrary number from 0-…., then why in figure 2, it moves to -2 scale.
: We appreciate the reviewer's detailed comment. The slight decrease in HFMSE score in patient 4 of figure 1 during the period of nusinersen administration is considered as a temporary deconditioning of the patient due to acute medical condition rather than a reversal of the effect of nusinersen. In fact, in the case of patient 4, the evaluator reported that the general condition was worse than usual at the time of the HFMSE examination (at 10 months after the start of the nusinersen administration) since the subject was admitted to the hospital with pneumonia one month before the test. The criteria for discontinuing nusinersen is limited to cases where maintenance or improvement of motor function cannot be demonstrated for 2 consecutive measurement because the SMA patient has a high risk of an acute medical condition such as respiratory infection and a temporary deterioration of physical function is frequent.
If the PP score is lower than the baseline test (prior to administration of nusinersen), it is marked as a minus since the y axis is defined as ‘change in test score from baseline’ in figure 2. In the case of Patient 3 in Figure 2, the PP score performed 1 month after the start of nusinersen administration (Preferred hand and Non-preferred hand, 3 points and 3 points) was temporarily lower than the baseline PP test score (5 points and 5 points), so it was marked as a minus value. In both patients, because the HFMSE and PP tests showed the tendency of improvement after nusinersen administration during long-term follow-up and there were no cases where maintenance or improvement of motor function cannot be demonstrated for 2 consecutive measurement, it is considered as the temporary decline of physical functions caused by an acute medical condition rather than regarding as reversal of effect of nusinersen.
Reference
- Verhaart, I.E.C.; Robertson, A.; Wilson, I.J.; Aartsma-Rus, A.; Cameron, S.; Jones, C.C.; Cook, S.F.; Lochmüller, H. Prevalence, incidence and carrier frequency of 5q-linked spinal muscular atrophy - a literature review. Orphanet J Rare Dis 2017, 12, 124, doi:10.1186/s13023-017-0671-8.
- Darras, B.T.; Chiriboga, C.A.; Iannaccone, S.T.; Swoboda, K.J.; Montes, J.; Mignon, L.; Xia, S.; Bennett, C.F.; Bishop, K.M.; Shefner, J.M.; et al. Nusinersen in later-onset spinal muscular atrophy: Long-term results from the phase 1/2 studies. Neurology 2019, 92, e2492-e2506, doi:10.1212/wnl.0000000000007527.
- De Vivo, D.C.; Bertini, E.; Swoboda, K.J.; Hwu, W.L.; Crawford, T.O.; Finkel, R.S.; Kirschner, J.; Kuntz, N.L.; Parsons, J.A.; Ryan, M.M.; et al. Nusinersen initiated in infants during the presymptomatic stage of spinal muscular atrophy: Interim efficacy and safety results from the Phase 2 NURTURE study. Neuromuscul Disord 2019, 29, 842-856, doi:10.1016/j.nmd.2019.09.007.
- Finkel, R.S.; Mercuri, E.; Darras, B.T.; Connolly, A.M.; Kuntz, N.L.; Kirschner, J.; Chiriboga, C.A.; Saito, K.; Servais, L.; Tizzano, E.; et al. Nusinersen versus Sham Control in Infantile-Onset Spinal Muscular Atrophy. N Engl J Med 2017, 377, 1723-1732, doi:10.1056/NEJMoa1702752.
- Mercuri, E.; Darras, B.T.; Chiriboga, C.A.; Day, J.W.; Campbell, C.; Connolly, A.M.; Iannaccone, S.T.; Kirschner, J.; Kuntz, N.L.; Saito, K.; et al. Nusinersen versus Sham Control in Later-Onset Spinal Muscular Atrophy. N Engl J Med 2018, 378, 625-635, doi:10.1056/NEJMoa1710504.
- Montes, J.; Dunaway Young, S.; Mazzone, E.S.; Pasternak, A.; Glanzman, A.M.; Finkel, R.S.; Darras, B.T.; Muntoni, F.; Mercuri, E.; De Vivo, D.C.; et al. Nusinersen improves walking distance and reduces fatigue in later-onset spinal muscular atrophy. Muscle Nerve 2019, 60, 409-414, doi:10.1002/mus.26633.
- Amirjani, N.; Ashworth, N.L.; Olson, J.L.; Morhart, M.; Chan, K.M. Validity and reliability of the Purdue Pegboard Test in carpal tunnel syndrome. Muscle Nerve 2011, 43, 171-177, doi:10.1002/mus.21856.
- Buddenberg, L.A.; Davis, C. Test-retest reliability of the Purdue Pegboard Test. Am J Occup Ther 2000, 54, 555-558, doi:10.5014/ajot.54.5.555.
- Causby, R.; Reed, L.; McDonnell, M.; Hillier, S. Use of objective psychomotor tests in health professionals. Percept Mot Skills 2014, 118, 765-804, doi:10.2466/25.27.PMS.118k27w2.
Reviewer 2 Report
Insufficient information about selection of these five patients, Were participant selected based on specific characterestics besides being SMA type 2? Why there are only females?
Methods should include statistical analysis.
Where there other characteristics of these patients such as tube feeding and respiratory support.
There was no comparative analysis done neither with unaffected children or affected without treatment (natural history)
There is significant discrepancy between gain of points in PP test of SMA patients vs non-SMA (historical data). This should be discussed and attempted to explain.
Author Response
Dear Reviewer,
We really appreciate the reviewer’s detailed comments on our manuscript. The reviews have raised some excellent questions, and we sought to address each issue in detail in this letter and in the attached revised manuscript. You will find the reviewer’s comments in italic, followed by our detailed response. The original text is shown in blue, and the revised or added text is shown in red.
We hope that you will find our responses to be adequate. We thank you again for your time and effort spent reading and commenting on our manuscript.
1) Insufficient information about selection of these five patients, Were participant selected based on specific characteristics besides being SMA type 2? Why there are only females?
: We appreciate the reviewer's detailed comment. In recruiting participants for this study, there was no intentional selection according to specific characteristics other than SMA. If SMA is confirmed through genetic diagnosis and SMA-related clinical symptoms develop in children under 3 years of age (and a permanent ventilator is not used), the national health insurance in Korea is supporting the treatment of nusinersen. The use of a permanent ventilator is also applied as an end-point of the use of nusinersen in other studies [1,2], and this study does not consider other special characteristics except for the condition that a permanent ventilator is not used. In the hospital where this study was conducted, although the researchers did not intend to, only female patients with SMA type 2 were subject to nusinersen and participated in the study. The part pointed out by the reviewer has been added to “2.1. Patients”.
[2.1. Patients 1st paragraph]
A total of five patients with 5q SMA confirmed based on SMN1 genetic documentation were included in this study. All patients had a clinical classification of SMA type 2.
à
A total of five patients with 5q SMA confirmed based on SMN1 genetic documentation were included in this study. All patients had a clinical classification of SMA type 2 and received neither permanent ventilator support nor enteral feeding.
2) Methods should include statistical analysis.
: We really appreciate the very important comment of the reviewer. Since this study is a case series study and there are only 5 subjects, this study is not satisfied with the minimum number of subjects required to perform the Wilcoxon signed rank test (non-parametric test) to compare statistically significant differences before and after nusinersen treatment. Nevertheless, it is considered that this study is meaningful as a first case report on the improvement of fine manual dexterity in SMA patients by administration of nusinersen. Although statistical analysis could not be performed, since the study population showed an improvement trend in fine manual dexterity through a long-term follow-up of 18 months, statistical significance of the effect of nusinersen on fine manual dexterity can also be verified through following additional multi-center and large-scale study.
3) Where there other characteristics of these patients such as tube feeding and respiratory support.
: We appreciate the reviewer's detailed comment. None of the subjects received tube feeding support, and there was no history of respiratory support such as tracheostomy or mechanical ventilation (including temporary support). These information have been added to “2.1. Patients”.
[2.1. Patients 1st paragraph]
A total of five patients with 5q SMA confirmed based on SMN1 genetic documentation were included in this study. All patients had a clinical classification of SMA type 2.
à
A total of five patients with 5q SMA confirmed based on SMN1 genetic documentation were included in this study. All patients had a clinical classification of SMA type 2 and received neither permanent ventilator support nor enteral feeding.
4) There was no comparative analysis done neither with unaffected children or affected without treatment (natural history)
: We really appreciate the very important comment of the reviewer. As the reviewer pointed out, if there was a control group consisting of unaffected children or untreated SMA type 2, it would have been possible to present a more definite result. In the manuscript that was initially submitted, only patient 2 was mentioned as an example for the comparison of the PP test between unaffected children and study participants, but data for all subjects was added as figure 3 according to the reviewer's comment. As shown in figure 3 which is newly added through revision, although most of the subjects in this study recorded PP test scores lower than the normal children of the same age and sex at baseline before the start of nusinersen treatment, the PP test scores of most subjects improved to near normal range compared with the healthy children of same age and sex after 18 months of treatment. Considering that the hand dexterity of untreated SMA patients is lower than that of normal people of the same age [3] and the natural course of SMA is in the direction of progressive deterioration of motor function [4], the improvement of fine manual dexterity in SMA type 2 patients shown in figure 3 is presumed due to the treatment of nusinersen. Comparisons with unaffected children are added to “3.2.2. Purdue Pegboard test”.
[3.2.2. Purdue Pegboard test 2nd paragraph]
The PP score of the non-preferred hand also improved in all patients, ranging from +3 to +7 (+60–500%) when compared to the baseline score (Table 3 and Figure 2).
à
The PP score of the non-preferred hand also improved in all patients, ranging from +3 to +7 (+60–500%) when compared to the baseline score (Table 3 and Figure 2b).
In Figure 3, most subjects—except for patient 3—had lower PP scores than normative data of the same age and sex before nusinersen administration. However, the PP scores in both hands of patients 2, 3, and 5 after nusinersen administration (at 18 months) improved to the normal range.
(a)
(b)
Figure 3. Comparison of Purdue Pegboard score changes in patients versus normative data. (a) Preferred hand; (b) Non-preferred hand.
5) There is significant discrepancy between gain of points in PP test of SMA patients vs non-SMA (historical data). This should be discussed and attempted to explain.
: We really appreciate the very important comment of the reviewer. As mentioned in above 4), all the SMA patients except for patient 3 had lower PP test scores than normal subjects of the same age and sex at baseline before receiving nusinersen, but the fine manual dexterity of both hands of patients 2, 3, and 5 measured by the PP test improved to within the normal range of healthy children of the same age and sex after 18 months of treatment (Figure 3). In a previous study, it was reported that when nusinersen was administered to infants at the pre-symptomatic stage who have not yet developed SMA symptoms, most participants achieved the motor milestone within the window for healthy children [1]. Similar to the results of the previous study, the results of this study are considered that the administration of nusinersen caused an improvement in fine manual dexterity in SMA patients and allowed them to follow normal motor milestones by rapidly narrowing the gap with healthy children. Texts related to this have been added to the discussion section.
[4. Discussion 4th paragraph]
In studies related to PP performance, improvements in PP score are commonly observed with age in both preschool and school-aged children. While this may have also accounted for some of the increase in PP scores observed in our patients, the natural course of SMA is generally associated with reduced motor function in the absence of nusinersen. Furthermore, the increase in PP score observed in our patients was larger than that reported for healthy children of the same age and sex. Therefore, it is more reasonable to regard the increase in PP score as an effect of nusinersen administration. For example, in patient 2, the PP score increased by 7 points in both the preferred hand and the non-preferred hand after 18 months after nusinersen administration. On the other hand, for healthy children of the same age and sex, the average PP score increases by 1.9 points in the preferred hand and 1.83 points in the non-preferred hand over the same period (18 months).
à
In comparing the fine manual dexterity of SMA type 2 patients treated with nusinersen versus normal children, most SMA patients before receiving nusinersen had lower PP test scores than normal children of the same age and sex. However, the scores improved to the normal range observed from healthy children in most patients treated for 18 months (Figure 3) [8,13]. In a previous study, it was reported that when nusinersen was administered to infants during the pre-symptomatic stage, most patients achieved the motor milestone within the window for healthy children [14]. Similar to the results of the previous study, it is postulated that nusinersen administration rapidly improved fine manual dexterity in SMA patients; thus, they were able to reduce the gap with the normal fine motor milestone.
Reference
- De Vivo, D.C.; Bertini, E.; Swoboda, K.J.; Hwu, W.L.; Crawford, T.O.; Finkel, R.S.; Kirschner, J.; Kuntz, N.L.; Parsons, J.A.; Ryan, M.M.; et al. Nusinersen initiated in infants during the presymptomatic stage of spinal muscular atrophy: Interim efficacy and safety results from the Phase 2 NURTURE study. Neuromuscul Disord 2019, 29, 842-856, doi:10.1016/j.nmd.2019.09.007.
- Finkel, R.S.; Mercuri, E.; Darras, B.T.; Connolly, A.M.; Kuntz, N.L.; Kirschner, J.; Chiriboga, C.A.; Saito, K.; Servais, L.; Tizzano, E.; et al. Nusinersen versus Sham Control in Infantile-Onset Spinal Muscular Atrophy. N Engl J Med 2017, 377, 1723-1732, doi:10.1056/NEJMoa1702752.
- Janssen, M.; Peeters, L.H.C.; de Groot, I.J.M. Quantitative description of upper extremity function and activity of people with spinal muscular atrophy. J Neuroeng Rehabil 2020, 17, 126, doi:10.1186/s12984-020-00757-4.
- Wolfe, A.; Scoto, M.; Milev, E.; Muni Lofra, R.; Abbott, L.; Wake, R.; Rohwer, A.; Main, M.; Baranello, G.; Mayhew, A.; et al. Longitudinal changes in respiratory and upper limb function in a pediatric type III spinal muscular atrophy cohort after loss of ambulation. Muscle Nerve 2021, 64, 545-551, doi:10.1002/mus.27404.
Round 2
Reviewer 2 Report
Revised version looks good.